**Data Availability Statement:** The dataset used in this study is available from the Mendeley database (DOI: 10.17632/vwzrw7j6rx.1).

**Funding:** The author(s) received no specific funding for this work.

# Stress and anxiety among physicians during the COVID-19 outbreak in the Iraqi Kurdistan Region: An online survey

**Banaz A. Saeed**[1], **Nazar P. Shabila**📷[2]*, **Aram Jalal Aziz**📷[3]

**1** Department of Psychiatry, College of Medicine, Hawler Medical University, Erbil, Kurdistan Region, Iraq, **2** Department of Community Medicine, College of Medicine, Hawler Medical University, Erbil, Kurdistan Region, Iraq, **3** Erbil Psychiatric Hospital, Erbil, Kurdistan Region, Iraq

* nazarshabila@gmail.com

## Abstract

### Background

During infectious disease outbreaks, healthcare workers are at high risk of infection, infecting others, and psychological distress. This study aimed to determine the prevalence of stress and anxiety in physicians during the COVID-19 outbreak in the Iraqi Kurdistan Region and assess their associated factors.

### Methods

This cross-sectional study was carried out in Erbil, Iraqi Kurdistan Region, from March 28 to April 15, 2020. An online self-administered survey questionnaire was used to collect data from physicians working in specialized COVID-19 centers and other healthcare facilities. The level of stress was measured based on the 10-items Perceived Stress Scale. The level of anxiety was measured based on the 7-item Generalized Anxiety Disorder scale.

### Results

A total of 370 participants responded to the perceived stress component of the survey, of whom 57 (15.4%) had low perceived stress, 249 (67.3%) had moderate stress, and 64 (17.3%) had high stress. Being female was significantly associated with having moderate/high stress (adjusted odds ratio (AOR) = 2.40 (95% CI 1.31–4.39)). A total of 201 participants responded to the generalized anxiety disorder component of the survey, of whom 19 (9.5%) had no anxiety, 57 (28.4%) had mild anxiety, 79 (39.3%) had moderate anxiety, and 46 (22.9%) had severe anxiety. Working in COVID-19 centers (AOR = 2.23 (95% CI 1.02–4.86)) and being general practitioners (AOR = 4.16 (95% CI 1.14–15.14)) were significantly associated with having moderate/severe anxiety.

### Conclusion

A considerable proportion of physicians experience stress and anxiety during the COVID-19 outbreak in Iraqi Kurdistan region. Generalists and those in special COVID-19 units report the greatest anxiety. There a need to establish mechanisms to reduce the risks of stress

**Competing interests:** The authors have declared that no competing interests exist.

and anxiety among physicians. Mental health coping interventions through counseling should be based on COVID-19 protocol guidelines. Interventions should also emphasize physicians' ability to work safely and efficiently in providing care to the patients.

## Introduction

Infectious diseases are considered one of the greatest threats to the well-being of people. They remain significant causes of mortality and morbidity, even with the significant advances in medicine. The new and reemerging infectious diseases continue to threaten people's health and well-being in both developed and developing countries [1].

The outbreak of severe acute respiratory syndrome coronavirus 2 (SARS-CoV-2) infection started in Wuhan, China, in December 2019. The disease caused by the virus is called the coronavirus disease 2019 (COVID-19). The World Health Organization soon announced the COVID-19 epidemic a public health emergency and a rapidly growing pandemic that has involved most countries around the world [2]. The primary mode of transmission of the virus is through person-to-person transmission, mostly by large airborne droplets. The virus is also transmitted through contact with contaminated surfaces [3].

Physicians, nurses, and other healthcare workers do great work in frontline and stressful settings every day. However, healthcare workers are also human beings and are liable to diseases. Even without the presence of major stressful situations such as epidemics and conflicts, healthcare workers still face growing stresses related to the nature of the work in the medical field [4].

The COVID-19 is considered a critical situation due to the considerably high transmission and mortality rates and lack of effective vaccine or treatment. The frontline healthcare workers are directly involved in providing care to patients with COVID-19 [5]. Besides the high risk of infection, healthcare workers are at risk of experiencing psychological anxiety, stress, and other mental health symptoms [6]. These healthcare workers are at high risk of mental health burden due to the continuously increasing number of COVID-19 cases and deaths, devastating workload, shortage of personal protection equipment, unavailability of specific and effective treatment or vaccines, and absence of adequate support [5, 6]. The widespread media coverage and the negative role of social media are also contributing to the risk of the mental health burden of healthcare workers [7]. Even healthcare workers who are not in the frontline with COVID-19 frequently contact non-diagnosed COVID-19 cases, as many cases are asymptomatic or have simple symptoms.

Adverse psychological reactions and mental health problems among healthcare workers have been reported during other similar infectious disease outbreaks, such as SARS [6, 8–10]. In these outbreak situations, healthcare workers will be affected and infected while providing care for patients. In 2003, healthcare workers comprised a substantial proportion of SARS victims [11]. In addition to the risk of infection during the 2003 SARS outbreak, healthcare workers were at the risk of stress and anxiety [8]. Studies of the SARS outbreaks in different countries revealed an enormous emotional burden among the frontlines healthcare workers, which ultimately resulted in psychological morbidity [1].

During previous similar outbreaks, healthcare workers were afraid of getting the infection and infecting their families. They also suffered from uncertainty and stigmatization, and many of them were reluctant to work and even resigned [6, 8]. They also experienced high levels of stress, depression, and anxiety symptoms with long-term psychological consequences [9]. Research has shown that the COVID-19 situation is associated with similar concerns for

frontline healthcare workers in terms of general well-being, psychological impact, and mental health symptoms [5, 12–14]. Healthcare workers are at high risk of workplace stress, although workplace stress occurs in all professions. Worldwide, around one-third of employees suffer from workplace stress. The unique work environment makes healthcare workers significantly impacted by workplace stress [15, 16].

In many developed countries, health authorities and mental health institutions have widely deployed psychological assistance and counseling services in response to the COVID-19 outbreak [5, 17]. However, these types of essential services are absent or limited in most developing countries, including Iraq. Moreover, there are a limited number of mental health interventions and evaluations that have targeted healthcare workers in frontline settings.

Research is limited about stress and anxiety among healthcare workers in the Iraqi context, particularly during the COVID-19 outbreak. This study aimed to determine the prevalence of stress and anxiety in physicians working in specialized COVID-19 centers and other general hospitals and health centers during the outbreak of COVID-19 in the Iraqi Kurdistan Region and assess their associated factors.

## Materials and methods

### Study setting

This cross-sectional study was carried out in Erbil, Iraqi Kurdistan Region, from March 28 to April 15, 2020. During this period, 337 cases of COVID-19 with four deaths were recorded in the region, and 7695 people were in quarantine. Many physicians were working or were on call in all hospitals and places specified for COVID-19 cases and quarantine. There were four COVID-19 specialized centers in Erbil governorate. Generally, there was a limited number of cases at the time of the survey, and most of them were asymptomatic, while only a few deaths were recorded. Asymptomatic cases constituted most diagnosed cases as they were diagnosed after being tested for being contacts of positive cases or having returned recently from abroad. There was a lack of personal protective equipment in most health institutions that are not specified for COVID-19 cases, such as primary health care centers. This was the first large epidemic in the Iraqi Kurdistan Region, and the provision of personal protective equipment to the health facilities has not been given a priority in the past.

### Study design and participants

An online self-administered survey questionnaire based on Google form was designed for data collection. The participants included the physicians working in the four specialized COVID-19 centers and other health care centers and hospitals in Erbil governorate. The online survey was shared with physicians by contacting the different hospitals and different professional specialty associations and sharing it in specific social media groups. The sample size was calculated using the Epi-info, assuming that the prevalence of stress among physicians in the COVID-19 pandemic context is 71.5% based on a previous study from China [5]. We found that a sample size of 417 physicians was sufficient to achieve a 95% confidence interval for the prevalence with ±3.5% precision. A more conservative and lower precision than the conventional 5% was chosen as the expected prevalence of stress was estimated to be within the upper 30% level (71.5%). The sample size was increased to 450 to account for non-response. Thus, the survey was shared with a total of 450 physicians who work in Erbil. A convenience sample of physicians was selected with efforts made to include physicians from all the COVID-19 centers, main hospitals, and most of the main primary health care centers throughout Erbil governorate. Finally, the participating physicians represented all the four COVID-19 centers, all the 27 public hospitals, and 21 main primary health care centers with a physician in Erbil

governorate. The selected 21 primary health care centers represented around 50% of all main primary health care centers in Erbil governorate. Since the survey was in the English language and it included specific medical terms, the possibility of having respondents from outside the target population, i.e., physicians, was very limited. Moreover, the survey was distributed either directly or through closed social media groups related to physicians. The complete set of data was also checked for the occupation and place of work fields, and it was confirmed that all participants were physicians. Physicians with COVID-19 infection and illness during the survey and those critically ill or with known mental disorders were excluded from the study.

## Study tools

The survey questionnaire included three main parts. The first part of the questionnaire included questions on the participants' demographic and professional characteristics, including occupation, sex (male or female), age, marital status, professional title, and workplace. The second and third parts of the questionnaire included the 10-items Perceived Stress Scale and the 7-item Generalized Anxiety Disorder (GAD-7) scale, respectively.

The level of stress was measured based on the 10-items Perceived Stress Scale, which comprised the second part of the survey questionnaire. The Perceived Stress Scale is a 10-question tool used to measure an individual's perception of stress in the past month based on a 5 point Likert scale [18]. The Perceived Stress Scale is a validated, commonly used, and easy-to-use stress questionnaire. The scale has established acceptable psychometric properties [19]. The physicians were asked about their thoughts and feelings over the last month. The participants were asked to select the frequency of feeling or thinking a certain way from never to very often (0 = never, 1 = almost never, 2 = sometimes, 3 = fairly often, and 4 = very often). We calculated the total perceived stress scale score by summation of the scores of each question. The higher the scores meaning, the greater the levels of stress. For the four positively stated items (items 4, 5, 7, and 8), the perceived stress scale scores were obtained by reverse-scoring the responses (i.e., 0 = 4, 1 = 3, 2 = 2, 3 = 1, and 4 = 0). The total sum score of the perceived stress scale can range from 0 to 40. The total scores of this measurement were interpreted as follows: 0–13 scores were considered low stress, 14–26 scores were considered moderate stress, and 27–40 scores were considered high perceived stress. Such interpretation was based on the study tools guidelines [18] and some previous similar studies from different contexts that used similar scores [20–22]. The validity and applicability of the perceived stress component of the questionnaire showed an internal consistency (Cronbach's alpha) of 0.82 and a reliability coefficient of 0.70.

The level of anxiety was measured based on the 7-item Generalized Anxiety Disorder (GAD-7) scale [23], which comprised the third part of the survey questionnaire. The scale has established acceptable psychometric properties [24, 25]. The physicians were asked to respond to seven questions about the level of anxiety over the past two weeks based on a 4 point Likert scale (0 = not at all, 1 = several days, 2 = over half a day, 3 = nearly every day. The total sum score of GAD-7 can range from 0 to 21. The total scores of this measurement were interpreted as follows: 0–4 scores were considered having no anxiety, 5–9 scores were considered mild anxiety, 10–14 scores were considered moderate anxiety, and 15–21 scores were considered severe anxiety. Such interpretation was based on the study tools guidelines [23]. The validity and applicability of the generalized anxiety disorder component of the questionnaire showed an internal consistency (Cronbach's alpha) of 0.80 and a reliability coefficient of 0.72.

The individual items in the stress and the generalized anxiety disorder instruments represent the severity in the individual items (10 items for stress and 7 items for anxiety). On the other hand, the scale scores reflect the frequency of symptoms (0–4 and 0–3, respectively). It should be noted that all of these items might not occur at one time or another in each

individual. The severity and frequency of the stress or anxiety experienced are what distinguishes an individual from another.

Most of the physicians in Iraqi Kurdistan Region are Kurds and speak the Kurdish language. However, medical education in this region and Iraq is entirely in the English language. As the physicians are fluent in the English language, particularly the medical language, they prefer and find it easier to respond to medical and health surveys in the English language. Therefore, the questionnaire used in this study and the stress and anxiety measurement tools that were originally in the English language were not translated to the local Kurdish language, and the response was in the English language. As most of the questions in the survey were set as required questions, particularly those questions from the two scales, we did not have partial or incomplete responses.

Before data collection, official written ethical approval was obtained from the Research Ethics Committee of the College of Medicine, Hawler Medical University (reference 8/7 dated March 23, 2020). The participants were requested to provide a written, online recorded informed consent before completing the survey. The survey was anonymous, and confidentiality of information was maintained.

## Statistical analysis

The statistical package for the social sciences (SPSS, version 23) was used for data entry and analysis. Moderate and high stress groups were combined for comparison with low stress. Dependent variables were dichotomized to have logistic regression analysis. Thus, normal and mild anxiety groups were combined, and moderate and severe anxiety were combined. The Chi-square test of association was used to compare proportions. Using Bonferroni correction to account for using six comparisons simultaneously in the Chi-square test of association, a conservative lower P value of $\leq 0.008$ was regarded as statistically significant (0.05/6 = 0.008). Univariate analysis was used to assess the association of perceived stress and anxiety with demographic and professional factors. Multivariable logistic regression was also used to control for the demographic and professional factors of the participants. Variables included in the multivariable logistic regression were selected based on the bivariate associations with the dependent variables. Odds ratios (ORs) and 95% confidence intervals were calculated. Crude odds ratio (COR) and adjusted odds ratio (AOR) were reported. COR is obtained from univariate analysis when considering the effect of only one predictor variable. AOR is obtained from multivariable logistic regression, and it represents the value that has been adjusted for the other covariates, including confounders. The binary outcome for logistic regression included low stress/moderate to high stress for the perceived stress component and normal to mild anxiety/moderate to severe anxiety for the generalized anxiety disorder component.

## Results

A total of 370 participants responded to the perceived stress component of the survey with a response rate of 82.2%. The mean±SD age of the participants was 31.0±6.89 years (range 23–63 years). Most participants were females (71.4%), $\leq$ 30 years old (57.3%), married (56.5%), junior house officers (33.5%), and working in the city center (83.8%). Sixty six (17.8%) participants were working in COVID-19 treatment centers. A total of 201 participants responded to the generalized anxiety disorder component of the survey with a response rate of 44.7%. The mean±SD age of the participants was 32.3±8.51 years (range 22–63 years). Most participants were females (68.2%), $\leq$30 years old (55.2%), married (61.2%), junior house officers (33.3%), and working in the city center (88.6%). Forty six (22.9%) participants were working in COVID-19 treatment centers (Table 1).

**Table 1. Demographic and professional characteristics of the participants.**

| Variables | Perceived stress component | | Generalized anxiety disorder component | |
|---|---|---|---|---|
| | **No.** | **(%)** | **No.** | **(%)** |
| **Gender** | | | | |
| Male | 106 | (28.6) | 64 | (31.8) |
| Female | 264 | (71.4) | 137 | (68.2) |
| **Age group** | | | | |
| ≤30 | 212 | (57.3) | 111 | (55.2) |
| 31–40 | 116 | (31.4) | 49 | 24.4) |
| >40 | 42 | (11.4) | 41 | (20.4) |
| **Marital** | | | | |
| Single | 161 | (43.5) | 78 | (38.8) |
| Married | 209 | (56.5) | 123 | (61.2) |
| **Place of work** | | | | |
| City center | 310 | (83.8) | 178 | (88.6) |
| Outside city center | 60 | (16.2) | 23 | (11.4) |
| **Work at COVID-19 center** | | | | |
| No | 304 | (82.2) | 155 | (77.1) |
| Yes | 66 | (17.8) | 46 | (22.9) |
| **Job title** | | | | |
| Junior house office | 124 | (33.5) | 67 | (33.3) |
| General practitioner | 72 | (19.5) | 30 | (14.9) |
| Senior house office | 99 | (26.8) | 51 | (25.4) |
| Specialist | 75 | (20.3) | 53 | (26.4) |
| **Total** | **370** | **(100.0)** | **201** | **(100.0)** |

Of the 370 participants, 57 (15.4%) had low perceived stress, 249 (67.3%) had moderate stress, and 64 (17.3%) had high stress. The mean±SD perceived stress score of the participants was 20.5±6.9. Table 2 shows the responses to the perceived stress scale. During the last month preceding the survey, 44.3% of the participants fairly often or very often felt upset because of unexpected events, 34.6% felt unable to control important things in their lives, 57.0% felt nervous and stressed, 35.9% could not cope with all the things that they do, 50% had been angered because of things that were outside their control, and 34.6% felt that difficulties were piling up so high that they could not overcome.

**Table 2. Physicians' response to perceived stress scale.**

| In the last month, how often | Never | Almost never | Sometimes | Fairly often | Very often |
|---|---|---|---|---|---|
| | No. (%) | No. (%) | No. (%) | No. (%) | No. (%) |
| have you been upset because of something that happened unexpectedly? | 23 (6.2) | 65 (17.6) | 118 (31.9) | 93 (25.1) | 71 (19.2) |
| have you felt that you were unable to control the important things in your life? | 47 (12.7) | 83 (22.4) | 112 (30.3) | 62 (16.8) | 66 (17.8) |
| have you felt nervous and "stressed"? | 15 (4.1) | 40 (10.8) | 104 (28.1) | 111 (30.0) | 100 (27.0) |
| have you felt confident about your ability to handle your personal problems? | 72 (19.5) | 132 (35.7) | 103 (27.8) | 51 (13.8) | 12 (3.2) |
| how often have you felt that things were going your way? | 27 (7.3) | 88 (23.8) | 157 (42.4) | 65 (17.6) | 33 (8.9) |
| have you found that you could not cope with all the things that you had to do? | 44 (11.9) | 84 (22.7) | 109 (29.5) | 84 (22.7) | 49 (13.2) |
| have you been able to control irritations in your life? | 41 (11.1) | 139 (37.6) | 127 (34.3) | 48 (13.0) | 15 (4.1) |
| have you felt that you were on top of things? | 34 (9.2) | 91 (24.6) | 137 (37.0) | 79 (21.4) | 29 (7.8) |
| have you been angered because of things that were outside of your control? | 24 (6.5) | 63 (17.0) | 98 (26.5) | 110 (29.7) | 75 (20.3) |
| have you felt difficulties were piling up so high that you could not overcome them? | 40 (10.8) | 78 (21.1) | 124 (33.5) | 87 (23.5) | 41 (11.1) |

Table 3 shows the prevalence of different degrees of stress among the participants according to their demographic and professional characteristics. High stress was more prevalent among female participants (20.5%) than male participants (9.4%), P = 0.003. High stress was also more prevalent among younger age participants, married, those working in the city center and COVID-19 centers, and the junior house officers, but not to a statistically significant degree.

When moderate and high stress were combined and compared with low stress, the prevalence of moderate/high stress was significantly higher among female participants (87.9%) than male participants (76.4%), P = 0.006. On multivariable logistic regression, only being female was significantly associated with having moderate/high stress (AOR = 2.40 (95% CI 1.31–4.39)) as shown in Table 4.

Of the 201 participants who responded to the anxiety component of the survey, 19 (9.5%) reported no anxiety, 57 (28.4%) had mild anxiety, 79 (39.3%) had moderate anxiety, and 46 (22.9%) had severe anxiety. The mean±SD anxiety score of the participants was 9.8±5.1. Table 5 shows the responses to the anxiety scale. During the last two weeks preceding the survey, 46.3% of the participants frequently ("over half the days" or "nearly every day") felt nervous, anxious, or on edge, 41.3% were not able to stop or control worrying, 53.2% were worrying too much about different things, 40.3% had trouble relaxing, and 46.3% felt afraid as if something awful might happen.

Table 6 shows the prevalence of different degrees of anxiety among the participants according to their demographic and professional characteristics. No significant statistical association was detected between anxiety and any of the variables.

When moderate and severe anxiety were combined and compared with the no anxiety and mild anxiety combined, no statistically significant association was detected between anxiety

**Table 3. Prevalence of different degrees of stress according to the demographic and professional characteristics of the participants.**

| Variable | Stress | | | | | | Chi square | P value |
|---|---|---|---|---|---|---|---|---|
| | Low | | Moderate | | High | | | |
| | No. | (%) | No. | (%) | No. | (%) | | |
| **Age group (years)** | | | | | | | | |
| ≤30 | 29 | (13.7) | 145 | (68.4) | 38 | (17.9) | 2.828 | 0.587 |
| 31–40 | 22 | (19.0) | 73 | (62.9) | 21 | (18.1) | | |
| >40 | 6 | (14.3) | 31 | (73.8) | 5 | (11.9) | | |
| **Gender** | | | | | | | | |
| Male | 25 | (23.6) | 71 | (67.0) | 10 | (9.4) | 11.765 | 0.003 |
| Female | 32 | (12.1) | 178 | (67.4) | 54 | (20.5) | | |
| **Marital** | | | | | | | | |
| Single | 21 | (13.0) | 114 | (70.8) | 26 | (16.1) | 1.771 | 0.412 |
| Married | 36 | (17.2) | 135 | (64.6) | 38 | (18.2) | | |
| **Place of work** | | | | | | | | |
| City center | 47 | (15.2) | 208 | (67.1) | 55 | (17.7) | 0.304 | 0.859 |
| Outside the city center | 10 | (16.7) | 41 | (68.3) | 9 | (15.0) | | |
| **Work at COVID-19 center** | | | | | | | | |
| No | 48 | (15.8) | 204 | (67.1) | 52 | (17.1) | 0.209 | 0.901 |
| Yes | 9 | (13.6) | 45 | 68.2) | 12 | (18.2) | | |
| **Job title** | | | | | | | | |
| Junior house office | 15 | (12.1) | 83 | (66.9) | 26 | (21.0) | 5.256 | 0.511 |
| General practitioner | 10 | (13.9) | 52 | (72.2) | 10 | (13.9) | | |
| Senior house office | 16 | (16.2) | 65 | (65.7) | 18 | (18.2) | | |
| Specialist | 16 | (21.3) | 49 | 65.3) | 10 | (13.3) | | |

**Table 4. Factors associated with perceived stress among physicians.**

| Variable | Stress | | | | Chi square | P value | COR (95%CI) | AOR (95%CI)* |
|---|---|---|---|---|---|---|---|---|
| | Low | | Moderate / high | | | | | |
| | No. | (%) | No. | (%) | | | | |
| **Age group (years)** | | | | | | | | |
| ≤30 | 29 | (13.7) | 183 | (86.3) | 1.653 | 0.438 | 1 | 1 |
| 31–40 | 22 | (19.0) | 94 | (81.0) | | | 0.68 (0.37–1.24) | 0.99 (0.42–2.34) |
| >40 | 6 | (14.3) | 36 | (85.7) | | | 0.95 (0.37–2.46) | 1.64 (0.43–6.25) |
| **Gender** | | | | | | | | |
| Male | 25 | (23.6) | 81 | (76.4) | 7.627 | 0.006 | 1 | 1 |
| Female | 32 | (12.1) | 232 | (87.9) | | | 2.24 (1.25–4.00) | 2.40 (1.31–4.39) |
| **Marital status** | | | | | | | | |
| Single | 21 | (13.0) | 140 | (87.0) | 1.220 | 0.269 | 1 | 1 |
| Married | 36 | (17.2) | 173 | (82.8) | | | 0.72 (0.40–1.29) | 0.92 (0.46–1.85) |
| **Place of work** | | | | | | | | |
| City center | 47 | (15.2) | 263 | (84.8) | 0.087 | 0.767 | 1 | 1 |
| Outside city center | 10 | (16.7) | 50 | (83.3) | | | 0.77 (0.42–1.89) | 0.98 (0.43–2.10) |
| **Work at COVID-19 center** | | | | | | | | |
| No | 48 | (15.8) | 256 | (84.2) | 0.193 | 0.661 | 1 | 1 |
| Yes | 9 | (13.6) | 57 | (86.4) | | | 1.19 (0.55–2.56) | 1.00 (0.44–2.28) |
| **Job title** | | | | | | | | |
| Junior house office | 15 | (12.1) | 109 | (87.9) | 3.234 | 0.357 | 1 | 1 |
| General practitioner | 10 | (13.9) | 62 | (86.1) | | | 0.85 (0.36–2.01) | 0.79 (0.31–2.03) |
| Senior house office | 16 | (16.2) | 83 | (83.8) | | | 0.71 (0.33–1.53) | 0.66 (0.27–1.62) |
| Specialist | 16 | (21.3) | 59 | (78.7) | | | 0.51 (0.23–1.10) | 0.37 (0.11–1.23) |

AOR adjusted odds ratio, CI confidence interval, COR crude odds ratio.

* Model fitting information: AIC 144.1, Likelihood ratio test P = 0.172, Goodness of fit P = 0.227, Pseudo R-square = 0.034.

and any of the characteristics. On multivariable logistic regression, only working in COVID-19 center (AOR = 2.23 (95%CI 1.02–4.86)) and being a general practitioner (AOR = 4.16 (95% CI 1.14–15.14)) were significantly associated with having moderate/severe anxiety as shown in Table 7.

## Discussion

This study assessed the prevalence and severity of stress and anxiety among the physicians in Erbil governorate at the early stage of the COVID-19 outbreak in the Iraqi Kurdistan Region.

**Table 5. Participants' response to the generalized anxiety disorder scale.**

| Item | Not at all | Several days | Over half the days | Nearly everyday |
|---|---|---|---|---|
| | No. (%) | No. (%) | No. (%) | No. (%) |
| 1. Feeling nervous, anxious, or on edge | 11 (5.5) | 97 (48.3) | 54 (26.9) | 39 (19.4) |
| 2. Not being able to stop or control worrying | 38 1(8.9) | 80 (39.8) | 57 (28.4) | 26 (12.9) |
| 3. Worrying too much about different things | 25 (12.44) | 69 (34.33) | 68 (33.83) | 39 (19.40) |
| 4. Trouble relaxing | 34 (16.9) | 86 (42.8) | 47 (23.4) | 34 (16.9) |
| 5. Being so restless that it's hard to sit still | 76 (37.8) | 73 (36.3) | 39 (19.4) | 13 (6.5) |
| 6. Becoming easily annoyed or irritable | 40 (19.9) | 82 (40.8) | 47 (23.4) | 32 (15.9) |
| 7. Feeling afraid as if something awful might happen | 29 (14.4) | 79 (39.3) | 58 (28.9) | 35 (17.4) |

**Table 6. Prevalence of different degrees of anxiety according to the demographic and professional characteristics of the participants.**

| Factor | Anxiety | | | | | | | | Chi square | P value |
|---|---|---|---|---|---|---|---|---|---|---|
| | Normal | | Mild | | Moderate | | Severe | | | |
| | No. | (%) | No. | (%) | No. | (%) | No. | (%) | | |
| **Gender** | | | | | | | | | | |
| Male | 10 | (15.6) | 18 | (28.1) | 22 | (34.4) | 14 | (21.9) | | 0.221 |
| Female | 9 | (6.6) | 39 | (28.5) | 57 | (41.6) | 32 | (23.4) | 4.408 | |
| **Age group (years)** | | | | | | | | | | |
| ≤30 | 8 | (7.2) | 31 | (27.9) | 43 | (38.7) | 29 | (26.1) | | 0.615 |
| 31–40 | 5 | (10.2) | 12 | (24.5) | 22 | (44.9) | 10 | (20.4) | 4.459 | |
| >40 | 6 | (14.6) | 14 | (34.1) | 14 | (34.1) | 7 | (17.1) | | |
| **Marital** | | | | | | | | | | |
| Single | 3 | (3.8) | 28 | (35.9) | 34 | (43.6) | 13 | (16.7) | | 0.023 |
| Married | 16 | (13.0) | 29 | (23.6) | 45 | (36.6) | 33 | (26.8) | 9.543 | |
| **Place of work** | | | | | | | | | | |
| City center | 16 | (9.0) | 53 | (29.8) | 70 | (39.3) | 39 | (21.9) | | 0.551 |
| Outside the city center | 3 | (13.0) | 4 | (17.4) | 9 | (39.1) | 7 | (30.4) | 2.103 | |
| **Work at COVID-19 center** | | | | | | | | | | |
| No | 19 | (12.3) | 44 | (28.4) | 58 | (37.4) | 34 | (21.9) | | 0.089 |
| Yes | 0 | (0.0) | 13 | (28.3) | 21 | (45.7) | 12 | (26.1) | 6.518 | |
| **Job title** | | | | | | | | | | |
| Junior house office | 4 | (6.0) | 22 | (32.8) | 27 | (40.3) | 14 | (20.9) | | 0.053 |
| General practitioner | 2 | (6.7) | 2 | (6.7) | 14 | (46.7) | 12 | (40.0) | 16.710 | |
| Senior house office | 4 | (7.8) | 16 | (31.4) | 22 | (43.1) | 9 | (17.6) | | |
| Specialist | 9 | (17.0) | 17 | (32.1) | 16 | (30.2) | 11 | (20.8) | | |

It also examined the main factors associated with stress and anxiety. In this study, the prevalence of stress was relatively high among the physicians who participated in this study, as 67.3% of the physicians had moderate stress, and 17.3% had high stress. Other studies have shown a very high stress level among healthcare workers during the COVID-19 pandemic. For example, a study from China showed similarly a very high rate of stress (71.5%) among healthcare workers exposed to COVID-19 [5].

The prevalence of anxiety was considerably high among the participants, as 39.3% had moderate anxiety, and 22.9% had severe anxiety. A lower level of anxiety was reported among healthcare workers in other settings. For example, a systematic analysis that included 12 studies revealed a pooled prevalence of anxiety of 23 2% [13], and a study from China showed a prevalence of 44.6% [5].

Mental and psychological symptoms have also been high in previous outbreaks of other infectious diseases. For example, a study from the acute SARS outbreak time revealed that 89% of frontline healthcare workers experienced psychological distress [10]. A considerably lower rate of anxiety has been reported in other non-infectious disease emergencies. A study from Saudi Arabia showed that 7.6% of the emergency healthcare workers in Saudi Arabia had severe anxiety [26].

Research evidence suggests that a high proportion of healthcare workers experience mental and psychological distress during the COVID-19 outbreak [13]. Another study showed that healthcare workers had experienced extensive strain due to the COVID-19 epidemic due to stress, anxiety, and depression symptoms. Severe symptoms were reported in 2.2–14.5% of the participants [12].

**Table 7. Factors associated with anxiety among physicians.**

| Variable | Anxiety level | | | | Chi square | P value | COR (95%CI) | AOR (95%CI) |
|---|---|---|---|---|---|---|---|---|
| | No/Mild | | Moderate / Severe | | | | | |
| | No. | (%) | No. | (%) | | | | |
| **Age groups (years)** | | | | | | | | |
| ≤30 | 39 | (35.1) | 72 | (64.9) | 2.639 | 0.267 | 1 | 1 |
| 31–40 | 17 | (34.7) | 32 | (65.3) | | | 1.02 (0.50–2.07) | 0.82 (0.27–2.44) |
| >40 | 20 | (48.8) | 21 | (51.2) | | | 0.57 (0.28–1.18) | 0.56 (0.13–2.45) |
| **Gender** | | | | | | | | |
| Male | 28 | (43.8) | 36 | (56.3) | 1.409 | 0.235 | 1 | 1 |
| Female | 48 | (35.0) | 89 | (65.0) | | | 1.44 (0.79–2.64) | 1.49 (0.77–2.86) |
| **Marital** | | | | | | | | |
| Single | 31 | (39.7) | 47 | (60.3) | 0.202 | 0.653 | 1 | 1 |
| Married | 45 | (36.6) | 78 | (63.4) | | | 1.14 (0.64–2.05) | 1.58 (0.72–3.50) |
| **Place of work** | | | | | | | | |
| City center | 69 | (38.8) | 109 | (61.2) | 0.601 | 0.438 | 1 | 1 |
| Outside the city center | 7 | (30.4) | 16 | (69.6) | | | 1.45 (0.57–3.70) | 1.12 (0.41–3.36) |
| **Work at COVID-19 center** | | | | | | | | |
| No | 63 | (40.6) | 92 | (59.4) | 2.314 | 0.128 | 1 | 1 |
| Yes | 13 | (28.3) | 33 | (71.7) | | | 1.74 (0.85–3.56) | 2.23 (1.02–4.86) |
| **Job title** | | | | | | | | |
| Junior house office | 26 | (38.8) | 41 | (61.2) | | 0.014 | 1 | 1 |
| General practitioner | 4 | (13.3) | 26 | (86.7) | 10.566 | | 4.12 (1.29–13.17) | 4.16 (1.14–15.14) |
| Senior house office | 20 | (39.2) | 31 | (60.8) | | | 0.98 (0.47–2.07) | 0.95 (0.37–2.42) |
| Specialist | 26 | (49.1) | 27 | (50.9) | | | 0.67 (0.32–1.37) | 0.91 (0.22–3.80) |

AOR adjusted odds ratio, CI confidence interval, COR crude odds ratio.

* Model fitting information: AIC 126.6, Likelihood ratio test P = 0.037, Goodness of fit P = 0.124, Pseudo R-square = 0.085.

The psychological response of healthcare workers to an infectious disease outbreak is complicated. The distress can be attributed to different issues such as vulnerability feeling, loss of control, self-health concerns, the transmission of infection, the health of family and friends, changes in work, and isolation [27]. The COVID-19 might increase personal risk perception as it is human-to-human transmissible [28, 29] and is associated with high morbidity and death [30]. The pressures and concerns of healthcare workers might be related to the anticipated lack of supplies and the increasing number of COVID-19 cases [31].

In this study, high stress level was significantly more prevalent among female participants than male participants. High anxiety level was more prevalent among married physicians, physicians working in COVID-19 centers, and general practitioners than other physicians. Another COVID-19 related study from China showed that nurses, women, and frontline healthcare workers reported a greater level of stress, anxiety, and other mental health symptoms than other healthcare workers [5]. Similarly, a systematic review revealed gender and occupational differences, as female healthcare professionals and nurses exhibited higher levels of affective symptoms than males and physicians, respectively [13]. A review of six articles showed that gender, profession, age, workplace, department of work, and psychological factors were associated with higher levels of anxiety, stress, and other mental symptoms. Research has suggested that COVID-19 is considered an independent risk factor for stress in healthcare workers [14]. Another study showed that the severity of mental symptoms was determined by different factors, including age, gender, job, nature of work, and closeness to COVID-19 patients [12].

In other emergency conditions not related to COVID-19, healthcare workers were reported to have the highest anxiety, followed by physicians and nurses. Gender and older age group among health professionals were statistically significantly correlated with higher anxiety [26]. In normal, non-outbreak situations, the high stress level has been associated with younger and female consultant physicians, as a study from Saudi Arabia has shown [32].

In this study, the prevalence of anxiety was significantly higher among the physicians working in COVID-19 centers than other physicians, but the stress level did not differ significantly. The lack of significant difference in stress levels between these two groups could be because of the limited number of cases, symptomatic cases, and deaths in the region during the study period. This can also be related to the fact that all physicians can become in contact with infected individuals during the clinic or hospital work as people might be infected, and they are not diagnosed or are asymptomatic but can still transmit the infection. Another factor responsible for this could be the lack of personal protective equipment in most health institutions that are not specified for COVID-19 cases, such as primary health care centers.

The findings of the current study indicate a need to establish mechanisms to reduce the risks of stress and anxiety and employ mental health coping interventions. Physicians and other frontline healthcare workers can benefit from counseling based on COVID-19 protocol guidelines. Frontline healthcare workers can also benefit from stress management interventions as part of the preparation for future outbreaks. Interventions should also emphasize the ability of physicians to work safely and efficiently when providing care to patients. The results suggest that generalists and those working in COVID-19 centers have the greatest anxiety. A range of possible interventions can be taken that favor safety and efficiency. The generalists need to have triage for fever and other symptoms and/or exposure to COVID-19 by the clinic staff member before seeing the physician. Patients and physicians should use masks and consider physical distancing in the office. Personal protective equipment should be used if an examination or procedure is needed. For those working in COVID-19 centers, there is a need to follow personal protective protocol. There is a need to raise awareness that clinic co-workers have shown to be an important infection source in some settings compared to patients as a source of infection [33]. An important concern of utilizing a counseling/mental health approach at this stage is having potential alexithymic physicians who report minimal or no anxiety. Thus, they may not adhere to protocols and become COVID = 19 carriers or active cases [34].

This study has several limitations. The study's scope was limited as the participants were only from Erbil governorate, limiting the generalization of our findings to other regions of Iraq and Iraqi Kurdistan that might be less or more affected by the outbreak. This study was conducted at the early stage of the outbreak when the number of patients, symptomatic patients, and deaths was limited. Only physicians were included in this study, while other healthcare workers such as nurses can be at higher risk of infection due to more close and frequent contact with patients and working for longer hours [35, 36]. Moreover, the political instability and economic difficulty in the region during the study period can be considered stressors and might have affected physicians.

The low overall response rate of the current study can introduce bias into the results and might jeopardize the validity of the interpretations. Low response rates continue to be an important issue for online surveys [37]. The low overall response rate was likely due to the absence of follow-up procedures in this online survey. While follow-up procedures usually help in increasing the response rate in traditional surveys, their role and impact on online surveys are not well studied [38]. The response rate was much higher for the stress component of the survey (82.2%) than the generalized anxiety disorder component (44.7%). The survey was initially designed as two separate Google forms, one for the perceived stress component and

the other for the generalized anxiety component. Such a design is problematic and analyzing the two sets of data together limits the validity of the study. Designing the survey as one form with different sections and as one submission will help in avoiding this difference in response rate, although this might not increase the overall response rate. We could not conduct a correlation between stress scores and generalize anxiety disorder scores because of using two separate Google forms. Conducting such a correlation would have been interesting and useful.

When anxiety is studied longitudinally over time, the affected person might develop other comorbid stress and anxiety related disorders, including post-traumatic stress disorder, panic disorder, and behavioral and psychiatric disorders (e.g., substance use disorder, pathological gambling, and major depressive disorder). This changing clinical phenomenon would be an advantage in a longitudinal study, in which the changes can be described [39]. In the current cross-sectional study, we do not know how the anxiety experience might have disappeared, advanced to generalized anxiety disorder, or switched to another disorder. Some anxiety disorders, such as obsessive-compulsive disorder or phobias, may not begin with generalized anxiety disorder but rather appear without a generalized anxiety disorder entrée.

Since there is no disability/dysfunction marker(s) in this study, we do not know whether the physicians were experiencing normal-range emotions or symptoms related to disability and dysfunction. This absence of a distinction between normality and disability is a limitation to this study. In the absence of information on disability and dysfunction, these emotional experiences of stress or anxiety may be healthy under the circumstances of a pandemic [40]. Dysphoric experiences often compose a key motivating factor to change and adapt. Nevertheless, some degrees of anxiety may be useful and realistic, given the circumstances. Attempting to rid physicians of non-disabling anxiety may be counter-productive and undermine physicians' efforts to adjust to stressors and reduce anxiety.

Some people look for challenges that involve anxiety experiences, which they interpret as "exciting" (a positive valence) rather than "undesirable" (a negative valence) [41]. Some people choose a medical career for its challenge and excitement, as do people in various other challenging professions. Only a longitudinal study can determine whether these cross-section generalized anxiety disorder ratings are normative for physicians in the Iraqi Kurdistan Region or reflect external stressors, such as the pandemic. It would have also been helpful if the physicians have reported their own subjective descriptors, concerns, or aspirations as they entered the pandemic experience at this early stage.

In this study, efforts were made to include physicians with different characteristics and from different settings. However, using a convenience sample rather than a random sample limits the generalizability of the findings to all the target population. Moreover, we could not conclude that the responders and non-responders were statistically similar in their background characteristics. This also limits the generalizability of the study findings.

## Conclusion

A relatively high proportion of physicians experience stress and anxiety during the COVID-19 outbreak in Iraqi Kurdistan region. The findings indicate that generalists and those in special COVID-19 units report the greatest anxiety. There a need to establish mechanisms to reduce the risks of stress and anxiety among physicians. Mental health coping interventions through counseling should be based on COVID-19 protocol guidelines. Interventions should also emphasize physicians' ability to work safely and efficiently in providing care to the patients. Such interventions include triage for symptoms and/or exposure to COVID-19 by clinic staff members before seeing a physician, ensuring the supply of personal protective equipment, and employing personal protective protocols.

## Author Contributions

**Conceptualization:** Banaz A. Saeed, Nazar P. Shabila, Aram Jalal Aziz.

**Data curation:** Banaz A. Saeed, Aram Jalal Aziz.

**Formal analysis:** Nazar P. Shabila.

**Methodology:** Banaz A. Saeed, Nazar P. Shabila.

**Writing – original draft:** Banaz A. Saeed, Nazar P. Shabila.

**Writing – review & editing:** Aram Jalal Aziz.

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
