## [Editor Report · Decision Letter 0]

8 Jan 2021

PONE-D-20-30168

Stress and anxiety among physicians during the COVID-19 outbreak in the Iraqi Kurdistan Region: An online survey

PLOS ONE

Dear Dr. Shabila,

Thank you for submitting your manuscript to PLOS ONE. After careful consideration, we feel that it has merit but does not fully meet PLOS ONE’s publication criteria as it currently stands. Therefore, we invite you to submit a revised version of the manuscript that addresses the points raised during the review process.

We look forward to receiving your revised manuscript.

Kind regards,

Joseph John Westermeyer

Academic Editor

PLOS ONE

2. In your Methods section, please provide a justification for the sample size used in your study, including any relevant power calculations (if applicable).

Furthermore, if the questionnaire you developed a questionnaire as part of this study and it is not under a copyright more restrictive than CC-BY, please include a copy, in both the original language and English, as Supporting Information.

Editor Comments:

Summary.

              This study from a district of Iraq could be considered representative of many countries facing the Covid-19 Pandemic.  At the time of the study there were a limited number of cases identified, and “only a few deaths.”  Most cases were asymptomatic; they were tested because of exposure to positive cases or recent international travel.  All study participants were physicians, from recent graduates in training to fully trained generalists and specialists.  Certain clinical units had been designated as COVID-19 units.  Personal protective equipment (PPE) was lacking in units not specified as serving COVID-19 cases. 

              A total of 370 physicians participated in the study and provided “part 1” information (i.e., gender, age, marital status, urban/non-urban practice, working at COVID-19 center versus working elsewhere, and job title).  “Part 2,” also completed by 370 people, consisted of the Perceived Stress Scale, a 10-item, 0-4 scale per item reflecting frequency (over last month) of experiencing manifestations of stress (i.e., feeling upset, unable to control own life, nervous and stressed, loss of self-confidence, things going against self, unable to cope, irritable, overwhelmed, etc.), 6 items phrased in the negative and 4 items phrased in the positive, and total possible scores from 0 to 40, validated, widely used self-administered questionnaire.  “Part 3,” completed by 201 physicians, consisted of the 7-item Generalized Anxiety Disorder (e.g., feeling nervous/on-edge, worrying, trouble relaxing, irritable, anticipating doom), 0-3 scale indicating frequency over last two weeks, total score from 0 to 28.  This scale has also been widely used and validated. 

              The raw demographic data are shown in Tables 1 (demography/professional characteristics for “stress” scale in one column (n = 370) “anxiety” in another column (n = 201).  Table 2 shows the distribution of stress items in 370 physicians, and Table 5  shows the distribution of anxiety items in 201 physicians.  Table 3 compares the six demographic-professional against three stress levels (low vs. moderate vs. high), to produce Chi Square scores and probabilities.  Table 4 compares the demographic-professional variables against two stress levels (low, moderate/high) using the 95% Confidence Interval plus a multivariate analysis probability.   Tables 6 and 7 (similar to tables 4 and 5) compare the six demographic-professional against anxiety, using 4 levels of anxiety in Table 6 and two levels of anxiety in Table 7. 

              The authors set the probability level for significance at .05.  Their observations included 12 Chi Square comparisons and 12 regression analyses, which showed  significant comparisons, of which 2 differences involved male vs. females (more stress and anxiety among women), and 1 regression difference (more anxiety among generalist physicians).  Among the 36 95% confidence interval analyses, both COR and AOR were higher for generalist physicians (ORs = 4.12 and 4.16).  AOR-only was higher for the COVID-19 center workers (AOR = 2.23).

              The authors conclude, “A considerable proportion of physicians experience stress and anxiety during the COVID-19 outbreak in Iraqi Kurdistan region.  There is a need to establish mechanisms to reduce the risks of stress and anxiety and employ mental health coping interventions.  Additional research is required in this field.”

Main claims and significance for the discipline.

              The perceived stressors among physicians, along with their mental health are important areas of research during this Covid-19 Pandemic.

              The Introduction covers the core issues well, contains an adequate number of relevant citations, and prepares the reader for the subsequent sections. (One exception: most of one paragraph on background in the Discussion should be moved to the Introduction or Methods section; this problem is described below.)

              Using the entire population of physicians in an entire region provides an appropriate sample.

              All of the physicians in the sample (n = 370) participated in Parts 1 and 2 of the study.

              The demographic-professional characteristics of the physicians are salient and might be compared or extrapolated to other national groups of physicians. 

              The instruments used to measure “stress” and the “anxiety symptoms associated with Generalized Anxiety Disorder” have been well studied, validated, and employed in numerous surveys of both patients and “normals” (i.e., clinical and non-clinical samples).

              The Chi Square and regression statistics were well selected to compare the association between physician characteristics on one hand, and “perceived stress” plus “GAD anxiety experiences” on the other hand.

              The 95% Confidence Intervals compare the epidemiological rates between the physician characteristics on one hand versus “perceived stress” and “GAD-anxiety experiences” on the other hand.

              The data collected, as well as the population studied are appropriate for self-administration of an on-line questionnaire.

              Relatively few demographic-professional variables are associated with stress (mainly generalists and those in COVID-19 centers) or with GAD-anxiety (mainly gender). 

              Potential interventions are considered (primarily mental health), but no data were obtained regarding interventions.

              The report reads well overall.

Claims are properly placed in the context of the previous literature, and the authors treated the literature fairly.

The literature utilizing self-rating of emotional/psychological information supports the importance of language and culture in questionnaires.  The authors should provide the following information to the readers:What was the original language of the stress and GAD-anxiety instruments?What language or languages were used in the questionnaire?  Were the language or languages the primary languages of the physicians?What were the primary languages of the 370 physicians in the study?If a translation was conducted, how was it done?  Specific information on the original translation, back-translation into the original language, number and language backgrounds of those in creating the translation and/or any pilot/feasibility study of the original translation should be added.If translations were used, were the translated versions of the “stress” and “GAD anxiety” instruments re-standardized?  Is so, were they re-standardized using a clinical sample and/or non-clinical sample?

Do the data and analyses fully support the claims?  What other evidence is required?

Using a GAD-based instrument to assess anxiety may be problematic in certain ways, i.e.:The authors have chosen only one of several manifestations of anxiety, i.e., Generalized Anxiety Disorder or GAD.  Why did they decide not to include other manifestations of anxiety, such as Posttraumatic Stress Disorder or PTSD (which occurs considerably more often than GAD in many stressed and most traumatized populations), specific phobias (which can be as common as GAD in stressful or traumatic settings), social phobia, panic disorder, and obsessive-compulsive disorder or OCD?  The absence of other anxiety-related experiences would comprise a limitation to a study having generic “anxiety” in the title (rather than “GAD-anxiety”).Generalized Anxiety Disorder or GAD, when studied longitudinally over time, may morph into other disorders, some of them anxiety disorders (e.g., PTSD, Panic) and some other behavioral and psychiatric disorders (e.g., Substance Use Disorder, Pathological Gambling, Major Depressive Disorder).  This changing clinical phenomenon would be an advantage in a longitudinal study, in which the changes can be described.  In a cross-sectional study (such as this one), it is a limitation because we do not know how the GAD-anxiety experiences might have disappeared, advanced to GAD, or switched to another disorder.  Some anxiety disorders, such as OCD or phobias, may not begin with GAD but rather appear without a GAD entrée.Since there is no disability/dysfunction marker(s) in this study, we do not know whether the physicians were experiencing normal-range emotions, or symptoms of a kind that accompany disability/dysfunction. This absence of a distinction between normality and disability is a limitation to the study.In the absence of information on disability/dysfunction, these emotional experiences of stress or anxiety may be healthy under the circumstances  of a Pandemic.  Dysphoric experiences often compose a key motivating factor to change and adapt.Stated differently, a modicum of anxiety may be useful and realistic, given the circumstances.  Attempting to rid physicians of non-disabling anxiety may be counter-productive and undermine physicians’ efforts at adjusting to stressors and thereby reducing anxiety.Some people seek out challenges that involve anxiety experiences (such as those in GAD), but which they interpret as “exciting” (a positive valence) rather than “undesirable” (a negative valence).  Some people chose a medical profession for its challenge and excitement, as do people in a variety of other challenging occupations.  Without a longitudinal study, we do not know whether these cross-section GAD ratings are normative for physicians in this province or reflect external stressors (such as the Pandemic).  It would have also been helpful if the physicians might report their own subjective descriptors or concerns or aspirations as they enter the pandemic experience at this early phase.  The latter absence is also a limitation.During the period of the study, were there political or insurgent stressors that might have affected physicians in this province of Iraq?The authors have made numerous statistical comparisons of associated variables at the .05 level.  For example, Tables 3, 4, 6, and 7 each have 6 comparisons.  Making numerous comparisons increases the rate of having a spurious comparison at .05 probability by chance alone.  The authors should consider using a correction factor to lower the level of probability.  For example, the Bonferroni correction (which equals .05/number of comparisons) would be .05/6 or .008 for the six comparisons in each of these tables.  Using this lower probability level would eliminate the significance of marital status in Table 6. The 95% Confidence Intervals are not affected in the same way as hypothesis-testing statistics, so that the 95% C.I.s observed in Tables 4 and 7 remain (i.e., females have higher OR’s than males for stress, females have higher OR’s than males for GAD-anxiety, working at a COVID-19 Center is associated with more GAD-anxiety on the AOR only, and generalists report more anxiety than three other physician groups on both the COR and the AOR). None of the data presented suggest to this reviewer that mental health interventions (e.g., counseling, anti-anxiety treatment) are warranted at this stage.  On the contrary, interventions should emphasize physicians’ ability to work safely and efficiently in the care of all their patients.  Their data indicate that generalists and those in special COVIC-19 units report the greatest GAD-anxiety.  Potential interventions that favor safety and efficiency might include:Generalists: triage for fever, symptoms, and/or exposure to COVID-19 by clinic staff member before seeing physician; patient and physician using mask and distancing in the office; using PPH equipment if exam or procedure is warranted (e.g., gloves, mask, possibly gown-cap-visor as needed).COVID-19 unit workers: need to follow PPH protocol to the letter; awareness that clinic co-workers have shown to be greater infection source in some settings as compared to patients as a source of infection.If the investigators decide eventually to utilize a counseling/mental health approach, I would be as concerned or more concerned about the potential alexithymic physicians who report minimal or no anxiety (and thus may not adhere to protocols and become COVID=19 carriers or active cases).  

Is the paper considered suitable for publication in its present form?

The conclusions (stated in the Abstract) that “a need to establish mechanisms to reduce the risks of stress and anxiety and employ mental health coping interventions” are not supported by the data, which show demographic-professional associations with stress and GAD anxiety.  Since no interventions have been studied, these conclusions should be deleted.  The statement regarding “Additional research” is unnecessary and trite; it should also be deleted.Abbreviations should be written out the first time they are used (e.g., COR, AOR).  Differences in computing the CORs and AORs should be specified.New background information is presented for the first time in the Discussion on page 17, from line 281 to line 291 (beginning with “This could be” and ending with “a priority in the past”).  This important contextual information should be moved from the Discussion to the Background or Methods sections.  (New data should not appear for the first time in the Discussion.)In several places the authors state that certain associations between demography-profession and stress and GAD-anxiety are “more prevalent” and then adding “not to a statistically significant degree.”  However, these associations did not meet the cut-off of .05 probability and therefore are not “more prevalent” by definition.  Thus, the following sentences should be deleted:Page 11, lines 192-194, “The prevalence was also higher….but not to a significant level.”Page 13, lines 210-214, “Severe anxiety was more prevalent…. but not to a significant level.”Page 14, lines 220-222, “The prevalence was also higher… but not to a significant level.”In the Discussion, on page 15, lines 232-233, the phrase “…the prevalence of stress was very high among the physicians…” is not clearly demonstrated by the data.  Considerably more data would be required to make this statement conclusively.I assume that the stress scores and the GAD-anxiety scores are correlated with each other.  However, conducting a correlation between the two would nonetheless be of interest.  First, are both scores normally distributed (so that a decision can be made regarding use of Pearson versus Spearman correlation)?  Second, is the correlation a strong, close one, or is it weaker and more distant?  Are there many case examples of high stress with low anxiety and/or low stress with high anxiety?  If mental health services might be warranted, it would be in those showing a reverse correlation between stress and GAD-anxiety.  For example, physicians with alexithmia in need of treatment might have high stress/low anxiety.  Those whose anxiety disorder has been tripped off by minimal stress might stand out by virtue of low stress/high anxiety.The accuracy of the sentence on page 17, lines 279-281 should be checked for accuracy; “In this study, the prevalence of stress was significantly higher among the physicians working in COVID-19 centers than other physicians, but the anxiety level did not differ significantly.  Table 4 shows no 95% C.I. difference in stress as a function of working in a COVID-19 center; and Table 7 reveals that those physicians working in a COVID-19 center reported more GAD-anxiety as measured by the adjusted odds ratio (AOR) of the 95% C.I.

Are details of the methodology sufficient to allow the experiments to be reproduced?

- Does the total population in the region amount to 370 at the time of the study?  This information is implied, but not clearly stated.  Was this the total number on a single day, or during 2.5 weeks of data collection?  For those trying to replicate this study in another setting, such details would be helpful.

- How were the authors able to elicit 100% participation in Part 1 and Part 2 data collection, but then lose 46% of participants in Part 3?  What might they have done to elicit greater participation in Part 3?  How important a limitation is the 46% loss of participants in Part 3? Were any additional data obtained?

- It appears that the individual items in the stress and for anxiety instruments tap into severity in the individual items (10 for stress, 7 for anxiety), and the scale scores (0-4 and 0-3 respectively) tap into frequency.  If so, such information would be helpful to readers, who are likely to consider that all of these items occur at one time or another in all human beings; but what distinguishes individuals from one another is the severity and frequency of the stress or anxiety experience.

- It is not clear how the authors determined what “stressor” or “GAD anxiety” scores would be assigned to various categories (such as “low-moderate-high” for stressors, and “not at all – several days – over half the days – nearly everyday” for GAD anxiety).  The distribution of the three stress levels in Table 3 seems to suggest that they employed a normal distribution since “moderate” percentages in the middle tended to contain about 2/3 of participants, whereas most “low” and “high” stressor groups contained about 1/6 of participants.  The GAD-anxiety distributions were similar, if “several days” and “over half of days” were merged into a single “middle” variable, which contained about 2/3 of most items. 

- What are the rationales for grouping selected subgroups into single groups, as was done in Tables 4 and 7 for example?

Is the manuscript well organized and written clearly enough to be accessible to non-specialists?

On page 12, line 200, the use of the English “normal” presents some problems – in part because “normal” can means different things in English.  The latter include “average” (a statistical concept), healthy or non-pathological (a health/wellness concept), socially or culturally acceptable, ethically or morally correct, etc.  Since 9.5% is clearly not in the middle of a normal distribution, it is not “statistically normal.”  It may or may not be psychologically “normal.”  One might consider that those who report no anxiety may be “normal” in a wellness/health or culturally-desirable sense.  However, it is possible that reporting no symptoms may be undesirable or unhealthy.  For example, this group may alexithymic (i.e., unable to perceive their own distress, or minimizing their distress, or ignoring their distress), schizoid, lying, in denial, not recognizing the seriousness of the situation, etc.  So, this small group may be abnormal in a variety of ways (i.e., statistical, health/wellness, most adaptive). In general, the manuscript is well organized and clearly written.
---

## [Author Response · Author response to Decision Letter 0]

19 Feb 2021

Dear Editor,

Thank you very much for sharing the editors and reviewers’ comments and suggestions on the manuscript entitled “Stress and generalized anxiety disorder among physicians during the COVID-19 outbreak in the Iraqi Kurdistan Region: An online survey.” We thank them for these valuable comments and suggestions. We have found them very useful to improve the quality and clarity of the manuscript.

We have made the necessary revision by responding to the suggested comments. Please find below explanations to the revision made through point-to-point response to the comments. All the changes are in red color text.

Thank you very much for considering the revised manuscript.

Best regards,

Comment

The authors should provide the following information to the readers:

• What was the original language of the stress and GAD-anxiety instruments?

• What language or languages were used in the questionnaire? Were the language or languages the primary languages of the physicians?

• What were the primary languages of the 370 physicians in the study?

• If a translation was conducted, how was it done? Specific information on the original translation, back-translation into the original language, number and language backgrounds of those in creating the translation and/or any pilot/feasibility study of the original translation should be added.

• If translations were used, were the translated versions of the “stress” and “GAD anxiety” instruments re-standardized? Is so, were they re-standardized using a clinical sample and/or non-clinical sample?

Author response:

Thank you very much for this comment. Details are now provided about using the original English language of the instruments (Page 8, 2nd paragraph). There was no need to translate the instruments into the local Kurdish language as it was easy for the physicians to respond to the English instruments. As their entire medical education was in the English language, it is easier for them to use the English language in these instruments related to their health conditions. 

Comment

Using a GAD-based instrument to assess anxiety may be problematic in certain ways, i.e.:

The authors have chosen only one of several manifestations of anxiety, i.e., Generalized Anxiety Disorder or GAD. Why did they decide not to include other manifestations of anxiety, such as Posttraumatic Stress Disorder or PTSD (which occurs considerably more often than GAD in many stressed and most traumatized populations), specific phobias (which can be as common as GAD in stressful or traumatic settings), social phobia, panic disorder, and obsessive-compulsive disorder or OCD? The absence of other anxiety-related experiences would comprise a limitation to a study having generic “anxiety” in the title (rather than “GAD-anxiety”).

Authors response:

Thank you for this important comment. This aspect of using only one manifestation of anxiety has been highlighted at the end of the discussion section and is considered a limitation to this study (Page 20, 2nd paragraph).

Comment

Generalized Anxiety Disorder or GAD, when studied longitudinally over time, may morph into other disorders, some of them anxiety disorders (e.g., PTSD, Panic) and some other behavioral and psychiatric disorders (e.g., Substance Use Disorder, Pathological Gambling, Major Depressive Disorder). This changing clinical phenomenon would be an advantage in a longitudinal study, in which the changes can be described. In a cross-sectional study (such as this one), it is a limitation because we do not know how the GAD-anxiety experiences might have disappeared, advanced to GAD, or switched to another disorder. Some anxiety disorders, such as OCD or phobias, may not begin with GAD but rather appear without a GAD entrée.

Authors response:

This limitation of a cross sectional study has been highlighted (Page 20, 3rd paragraph), and the advantages of a longitudinal study in this respect have been emphasized.

Comment

Since there is no disability/dysfunction marker(s) in this study, we do not know whether the physicians were experiencing normal-range emotions, or symptoms of a kind that accompany disability/dysfunction. 

• This absence of a distinction between normality and disability is a limitation to the study.

• In the absence of information on disability/dysfunction, these emotional experiences of stress or anxiety may be healthy under the circumstances of a Pandemic. Dysphoric experiences often compose a key motivating factor to change and adapt.

• Stated differently, a modicum of anxiety may be useful and realistic, given the circumstances. Attempting to rid physicians of non-disabling anxiety may be counter-productive and undermine physicians’ efforts at adjusting to stressors and thereby reducing anxiety.

Authors response:

This issue of lack of disability/dysfunction markers has been highlighted as a limitation of this study (End of page 20 and beginning of page 21), and its details are discussed thoroughly.

Comment

Some people seek out challenges that involve anxiety experiences (such as those in GAD), but which they interpret as “exciting” (a positive valence) rather than “undesirable” (a negative valence). Some people chose a medical profession for its challenge and excitement, as do people in a variety of other challenging occupations. Without a longitudinal study, we do not know whether these cross-section GAD ratings are normative for physicians in this province or reflect external stressors (such as the Pandemic). It would have also been helpful if the physicians might report their own subjective descriptors or concerns or aspirations as they enter the pandemic experience at this early phase. The latter absence is also a limitation.

Authors response:

Thank you very much for raising this important and relevant aspect of anxiety. This limitation of not being able to distinguish a normative response or presence of stressor has been discussed in detail (Page 21, 2nd paragraph).

Comment

During the period of the study, were there political or insurgent stressors that might have affected physicians in this province of Iraq?

Authors response:

Thank you. This is very much relevant to the Iraq Kurdistan Region situation. The influence of the political and economic situation during this period on the physicians has been discussed (Page 19, end of 2nd paragraph).

Comment

The authors have made numerous statistical comparisons of associated variables at the .05 level. For example, Tables 3, 4, 6, and 7 each have 6 comparisons. Making numerous comparisons increases the rate of having a spurious comparison at .05 probability by chance alone. The authors should consider using a correction factor to lower the level of probability. For example, the Bonferroni correction (which equals .05/number of comparisons) would be .05/6 or .008 for the six comparisons in each of these tables. Using this lower probability level would eliminate the significance of marital status in Table 6. 

Authors response:

The probability level for the Chi square test has been changed to .008, as suggested. The interpretation of the results has been changed accordingly (Page 8, 4th paragraph).

Comment

None of the data presented suggest to this reviewer that mental health interventions (e.g., counseling, anti-anxiety treatment) are warranted at this stage. On the contrary, interventions should emphasize physicians’ ability to work safely and efficiently in the care of all their patients. Their data indicate that generalists and those in special COVIC-19 units report the greatest GAD-anxiety. Potential interventions that favor safety and efficiency might include:

• Generalists: triage for fever, symptoms, and/or exposure to COVID-19 by clinic staff member before seeing physician; patient and physician using mask and distancing in the office; using PPH equipment if exam or procedure is warranted (e.g., gloves, mask, possibly gown-cap-visor as needed).

• COVID-19 unit workers: need to follow PPH protocol to the letter; awareness that clinic co-workers have shown to be greater infection source in some settings as compared to patients as a source of infection.

• If the investigators decide eventually to utilize a counseling/mental health approach, I would be as concerned or more concerned about the potential alexithymic physicians who report minimal or no anxiety (and thus may not adhere to protocols and become COVID=19 carriers or active cases). 

Authors response:

Thank you very much for this important suggestion. We completely agree with you. The issue of lack of enough justification for mental health interventions at this stage and the need for interventions that emphasize physicians' ability to work safely and efficiently has been discussed in detail (Page 18, last paragraph, and beginning of page 19).

Comment 

The conclusions (stated in the Abstract) that “a need to establish mechanisms to reduce the risks of stress and anxiety and employ mental health coping interventions” are not supported by the data, which show demographic-professional associations with stress and GAD anxiety. Since no interventions have been studied, these conclusions should be deleted. The statement regarding “Additional research” is unnecessary and trite; it should also be deleted.

Authors response:

Thank you very much for this useful suggestion. The conclusion has been re-written to consider this comment (in the Abstract and main conclusion on page 21).

Comment

Abbreviations should be written out the first time they are used (e.g., COR, AOR). Differences in computing the CORs and AORs should be specified.

Authors response:

The abbreviations are now written in full word the first time used (Abstract and page 9, 1st paragraph)

Comment

New background information is presented for the first time in the Discussion on page 17, from line 281 to line 291 (beginning with “This could be” and ending with “a priority in the past”). This important contextual information should be moved from the Discussion to the Background or Methods sections. (New data should not appear for the first time in the Discussion.)

Authors response:

Many thanks for this useful note. This new background information has been moved to the Methods section (First paragraph of page 6), and a short necessary part of it was left in the Discussion section (page 18, 2nd paragraph).

Comment

In several places the authors state that certain associations between demography-profession and stress and GAD-anxiety are “more prevalent” and then adding “not to a statistically significant degree.” However, these associations did not meet the cut-off of .05 probability and therefore are not “more prevalent” by definition. Thus, the following sentences should be deleted:

Page 11, lines 192-194, “The prevalence was also higher….but not to a significant level.”

Page 13, lines 210-214, “Severe anxiety was more prevalent…. but not to a significant level.”

Page 14, lines 220-222, “The prevalence was also higher… but not to a significant level.”

Authors response:

All this text describing the non-significant association was omitted.

Comment

In the Discussion, on page 15, lines 232-233, the phrase “…the prevalence of stress was very high among the physicians…” is not clearly demonstrated by the data. Considerably more data would be required to make this statement conclusively.

Authors response:

This sentence was revised to consider this comment (page 16, 1st paragraph).

Comment

I assume that the stress scores and the GAD-anxiety scores are correlated with each other. However, conducting a correlation between the two would nonetheless be of interest. First, are both scores normally distributed (so that a decision can be made regarding use of Pearson versus Spearman correlation)? Second, is the correlation a strong, close one, or is it weaker and more distant? Are there many case examples of high stress with low anxiety and/or low stress with high anxiety? If mental health services might be warranted, it would be in those showing a reverse correlation between stress and GAD-anxiety. For example, physicians with alexithmia in need of treatment might have high stress/low anxiety. Those whose anxiety disorder has been tripped off by minimal stress might stand out by virtue of low stress/high anxiety.

Authors response:

Thank you very much for this suggestion. We highlighted the limitation of not being able to conduct such an important correlation between stress scores and GAD scores (end of page 19 and beginning page 20) as two separate survey forms were used.

Comment

The accuracy of the sentence on page 17, lines 279-281 should be checked for accuracy; “In this study, the prevalence of stress was significantly higher among the physicians working in COVID-19 centers than other physicians, but the anxiety level did not differ significantly. Table 4 shows no 95% C.I. difference in stress as a function of working in a COVID-19 center; and Table 7 reveals that those physicians working in a COVID-19 center reported more GAD-anxiety as measured by the adjusted odds ratio (AOR) of the 95% C.I.

Authors response:

Thanks for this note. This sentence was revised and corrected to reflect the actual finding (page 18, 2nd paragraph).

Comment 

Does the total population in the region amount to 370 at the time of the study? This information is implied, but not clearly stated. Was this the total number on a single day, or during 2.5 weeks of data collection? For those trying to replicate this study in another setting, such details would be helpful.

Authors response:

Details about the physician population and sample are provided now (Page 6, 2nd paragraph).

Comment

How were the authors able to elicit 100% participation in Part 1 and Part 2 data collection, but then lose 46% of participants in Part 3? What might they have done to elicit greater participation in Part 3? How important a limitation is the 46% loss of participants in Part 3? Were any additional data obtained?

Authors response:

Details about the response rate and the difference in response rate between the two surveys are provided (Page 9, first paragraph under Results and Page 19, last paragraph in Discussion).

Comment

It appears that the individual items in the stress and for anxiety instruments tap into severity in the individual items (10 for stress, 7 for anxiety), and the scale scores (0-4 and 0-3 respectively) tap into frequency. If so, such information would be helpful to readers, who are likely to consider that all of these items occur at one time or another in all human beings; but what distinguishes individuals from one another is the severity and frequency of the stress or anxiety experience.

Authors response:

Thanks for helping with this useful clarification. Details of this aspect of the severity of symptoms and frequency are provided now (Page 7, last paragraph).

Comment

It is not clear how the authors determined what “stressor” or “GAD anxiety” scores would be assigned to various categories (such as “low-moderate-high” for stressors, and “not at all – several days – over half the days – nearly everyday” for GAD anxiety). The distribution of the three stress levels in Table 3 seems to suggest that they employed a normal distribution since “moderate” percentages in the middle tended to contain about 2/3 of participants, whereas most “low” and “high” stressor groups contained about 1/6 of participants. The GAD-anxiety distributions were similar, if “several days” and “over half of days” were merged into a single “middle” variable, which contained about 2/3 of most items. 

Authors response:

Thanks for helping with the description of this procedure. Details of using the normal distribution to categorize the groups are provided now in the Methods section (Page 8, 1st paragraph).

Comment

What are the rationales for grouping selected subgroups into single groups, as was done in Tables 4 and 7 for example?

Authors response:

The rationale for grouping each of the stress and anxiety into two groups only (dichotomous variables) is now provided (Page 8, 4th paragraph).

Comment

On page 12, line 200, the use of the English “normal” presents some problems – in part because “normal” can means different things in English. The latter include “average” (a statistical concept), healthy or non-pathological (a health/wellness concept), socially or culturally acceptable, ethically or morally correct, etc. Since 9.5% is clearly not in the middle of a normal distribution, it is not “statistically normal.” It may or may not be psychologically “normal.” One might consider that those who report no anxiety may be “normal” in a wellness/health or culturally-desirable sense. However, it is possible that reporting no symptoms may be undesirable or unhealthy. For example, this group may alexithymic (i.e., unable to perceive their own distress, or minimizing their distress, or ignoring their distress), schizoid, lying, in denial, not recognizing the seriousness of the situation, etc. So, this small group may be abnormal in a variety of ways (i.e., statistical, health/wellness, most adaptive). 

Authors response:

This description is now corrected (Page 13, 1st paragraph) to avoid the word normal and the confusion it might cause.

---

## [Decision Letter · Decision Letter 1]

19 Apr 2021

PONE-D-20-30168R1

Stress and generalized anxiety disorder among physicians during the COVID-19 outbreak in the Iraqi Kurdistan Region: An online survey

PLOS ONE

Dear Dr. Shabila,

Thank you for submitting your manuscript to PLOS ONE. After careful consideration, we feel that it has merit but does not fully meet PLOS ONE’s publication criteria as it currently stands. Therefore, we invite you to submit a revised version of the manuscript that addresses the points raised during the review process.

The authors have presented findings from their study on some aspects of mental health among the selected physician population of Iraq during the ongoing COVID-19 pandemic. The authors need to address all the comments appended below. The authors should go through their manuscript thoroughly to check for typographic errors and correct them.

We look forward to receiving your revised manuscript.

Kind regards,

Arista Lahiri, MBBS, MD

Academic Editor

PLOS ONE

Additional Editor Comments (if provided):

The authors have presented a study on the mental health status of the Iraqi participants during the COVID19 pandemic. Apart from the reviewers' comments I have the following comments for the authors:

1. What was the sampling design and technique? Based on this were the obtained results valid? Were the findings generalizable to the target population? How was the sample size calculated?

2. The study tools require elaboration in terms of their validity and reliability measures as observed in the current sample. This is required to understand the validity of the measurements.

3. Since this is an online survey, how did the authors control for response from outside the target population, because random social media share often presents with this difficulty, unless the questionnaire has control questions to restrict the survey among the target population only.

4. What was the response rate? Was response rate accounted when calculating the outcome measures?

5. How was partial/ incomplete response handled in the study?

6. Were the authors able to conclude that those responded and the non-responders were statistically similar in terms of their background characteristics? If yes, then this finding may be incorporated as a supplementary information. However, if not, then authors need to acknowledge this limitation as it again threatens the generalizability of the research.

I recommend the authors to follow appropriate reporting guidelines for online survey research and modify accordingly.

Optional recommendation:

The authors may consider dividing their methods section in some appropriate sub-headings. This will make the article even more comprehensive.

Reviewers' comments:

Reviewer's Responses to Questions

**Comments to the Author**

1. If the authors have adequately addressed your comments raised in a previous round of review and you feel that this manuscript is now acceptable for publication, you may indicate that here to bypass the “Comments to the Author” section, enter your conflict of interest statement in the “Confidential to Editor” section, and submit your "Accept" recommendation.

Reviewer #1: (No Response)

Reviewer #2: All comments have been addressed

2. Is the manuscript technically sound, and do the data support the conclusions?

Reviewer #1: Yes

Reviewer #2: Partly

3. Has the statistical analysis been performed appropriately and rigorously? 

Reviewer #1: Yes

Reviewer #2: No

4. Have the authors made all data underlying the findings in their manuscript fully available?

Reviewer #1: Yes

Reviewer #2: Yes

5. Is the manuscript presented in an intelligible fashion and written in standard English?

Reviewer #1: Yes

Reviewer #2: Yes

6. Review Comments to the Author

Reviewer #1: The article needs following clarifications:

1. It is stated that there are 4 COVID 19 specialized centers. Whether all these centers were covered? Again, how many health centers and hospitals were covered in the study? These selected centers and hospitals constitute what proportion of the entire study settings in Iraqi Kurdistan Region? It has to be specified, since the title conveys physicians among Iraqi Kurdistan Region.

2. Regarding PSS: “The total scores of this measurement were interpreted as follows: 0-13 scores were considered low stress, 14-26 scores were considered moderate stress, and 27-40 scores were considered high perceived stress” – what is the reference?

3. Regarding GAD7: “0-4 scores were considered having no anxiety, 5-9 scores were considered mild anxiety, 10-14 scores were considered moderate anxiety, and 15-21 scores were considered severe anxiety” – what is the reference?

4. “The survey questionnaire included three main parts” – it is mentioned that first part constitutes demographic variables and it is not specified about the second and third part? Are these PSS and GAD 7 respectively? Please specify.

5. Please mention about the - Ethics committee approval statement, reference number and date of approval.

6. “We employed a normal distribution to classify the participants into different categories of stress and anxiety based on their total scores in the two measurements. Stress was classified into three groups, while generalized anxiety disorder was classified into four groups” – on what basis three and four groups were created? Again, what is the reference?

7. “AOR represents the value that has been adjusted for the other covariates, including confounders” – how did you get it from univariate analysis? To my knowledge it will come from multivariable regression analysis. Please clarify.

8. “Normal and mild anxiety groups were combined, and moderate and severe anxiety were ............. with multiple logistics regression analysis” – delete the term multiple from here. Dependent variable was dichotomized only to have logistic regression analysis. Again the term would be ‘logistic’ instead of ‘logistics’.

9. Output of logistic regression is missing. What about model fitness, statistical significance of model, variation of dependent variables that can be explained from the independent variables etc.?

10. How independent variables were chosen in multivariable regression analysis?

11. The term multiple in logistic regression is misnomer. It would have been multivariable logistic regression instead of multiple logistic regression.

12. Table 1: Stress component: Age group & Job title – percentage is not coming to 100%. Please correct.

13. Table 3, 4, 6 & 7: Mention the chi-square statistic along with P value.

14. Table 4, 6, 7: Mention years within parenthesis beside age group.

15. Table 4, 7: Footnote: Correct as ‘Crude odds ratio’ instead of ‘Crudes odds ratio’.

16. Table 5: Item 1 – percentage is not coming to 100%. Also check the other percentages for any corrections.

17. Table 2 & 5: Better to mention the number and percentages as No. (%) format to have smooth look instead of keeping in separate columns.

Reviewer #2: Authors here did a study on Stress and anxiety symptoms among physicians during Covid19 in the Iraqi Kurdistan region. The topic is relevant; especially this area is under studied from middle east perspective.

However these issues should be addressed first-

1. Applying PSS and GAD in separate google forms and analyzing them together is problematic and lacks validity.

2. While using a scale, referencing a disorder as yes or no according to Normal distribution is not acceptable and flawed. Rather it should be validated first and then to be diagnosed as yes or no according the to the cutoff marks.

3. It was written as 'The level of generalized anxiety disorder was measured based on the 7-item Generalized Anxiety Disorder scale.' GAD7 is a screening tool for GAD and can also be used for measuring Anxiety symptoms. So reframing this sentence(similar sentences are repeated many times in the manuscript) should be done.

These things need major update and reanalysis should be done before further proceedings.

7. PLOS authors have the option to publish the peer review history of their article (what does this mean?). If published, this will include your full peer review and any attached files.

Reviewer #1: **Yes: **Indranil Saha

Reviewer #2: **Yes: **Seshadri Sekhar Chatterjee

---

## [Author Response · Author response to Decision Letter 1]

2 May 2021

Academic Editor Comments

1. What was the sampling design and technique? Based on this were the obtained results valid? Were the findings generalizable to the target population? How was the sample size calculated?

Authors’ response:

Thank you very much for this important comment. 

Details about the sampling design are provided (page 6 last paragraph and beginning of page 7).

Details about the sample size calculation are provide (page 6, last paragraph).

Details about the generalizability of the findings and the study limitation in this regard are provided (page 23, first full paragraph).

2. The study tools require elaboration in terms of their validity and reliability measures as observed in the current sample. This is required to understand the validity of the measurements.

Authors’ response:

Details about the validity and reliability measures of the study tools are provided (page 8, end of first paragraph and end of second paragraph).

3. Since this is an online survey, how did the authors control for response from outside the target population, because random social media share often presents with this difficulty, unless the questionnaire has control questions to restrict the survey among the target population only.

Authors’ response:

Thanks for this important note. Details about controlling for response from outside the target population are now provided (page 7, end of first paragraph).

4. What was the response rate? Was response rate accounted when calculating the outcome measures?

Authors’ response:

Details about the response rates are provided (page 10, first paragraph under Results section) and also it is emphasized in the limitations of the study (page 21, first full paragraph).

5. How was partial/ incomplete response handled in the study?

Authors’ response:

Details about avoiding partial or incomplete responses are provided (page 9 end of the first full paragraph).

6. Were the authors able to conclude that those responded and the non-responders were statistically similar in terms of their background characteristics? If yes, then this finding may be incorporated as a supplementary information. However, if not, then authors need to acknowledge this limitation as it again threatens the generalizability of the research.

Authors’ response:

This issue of inability to conclude that responders and non-responders have similar background characteristics is highlighted as a limitation of the study (page 23 end of the first full paragraph).

I recommend the authors to follow appropriate reporting guidelines for online survey research and modify accordingly.

Authors’ response:

The reporting guidelines are now followed as much as possible.

Optional recommendation:

The authors may consider dividing their methods section in some appropriate sub-headings. This will make the article even more comprehensive.

Authors’ response:

The methods section is now divided in to four sub-headings (pages 5-10).

Reviewer #1: 

1. It is stated that there are 4 COVID 19 specialized centers. Whether all these centers were covered? Again, how many health centers and hospitals were covered in the study? These selected centers and hospitals constitute what proportion of the entire study settings in Iraqi Kurdistan Region? It has to be specified, since the title conveys physicians among Iraqi Kurdistan Region.

Authors’ response:

Thank you very much for this important comment. Details about the inclusion of the four COVID-19 centers and the number of hospitals and health centers are provided (page 7, first paragraph).

2. Regarding PSS: “The total scores of this measurement were interpreted as follows: 0-13 scores were considered low stress, 14-26 scores were considered moderate stress, and 27-40 scores were considered high perceived stress” – what is the reference?

Authors’ response:

The reference for the interpretation of the PSS score is provided (page 8, first paragraph).

3. Regarding GAD7: “0-4 scores were considered having no anxiety, 5-9 scores were considered mild anxiety, 10-14 scores were considered moderate anxiety, and 15-21 scores were considered severe anxiety” – what is the reference?

Authors’ response:

The reference for the interpretation of GAD7 score is provided (page 8, second paragraph).

4. “The survey questionnaire included three main parts” – it is mentioned that first part constitutes demographic variables and it is not specified about the second and third part? Are these PSS and GAD 7 respectively? Please specify.

Authors’ response:

The three parts are now mentioned together (page 7, first paragraph under Study tools sub-heading).

5. Please mention about the - Ethics committee approval statement, reference number and date of approval.

Authors’ response:

The number and date of ethical approved is provided (page 9, second paragraph).

6. “We employed a normal distribution to classify the participants into different categories of stress and anxiety based on their total scores in the two measurements. Stress was classified into three groups, while generalized anxiety disorder was classified into four groups” – on what basis three and four groups were created? Again, what is the reference?

Authors’ response:

Thanks for raising this concern. The part about normal distribution classification is removed and the distribution is now based on the tool guidelines and previous studies. References for such distribution are provided (Page 8, first and second paragraphs). 

7. “AOR represents the value that has been adjusted for the other covariates, including confounders” – how did you get it from univariate analysis? To my knowledge it will come from multivariable regression analysis. Please clarify.

Authors’ response:

Thanks for this note. This was corrected (page 10, first paragraph).

8. “Normal and mild anxiety groups were combined, and moderate and severe anxiety were ............. with multiple logistics regression analysis” – delete the term multiple from here. Dependent variable was dichotomized only to have logistic regression analysis. Again the term would be ‘logistic’ instead of ‘logistics’.

Authors’ response:

Thanks for this correction. This was corrected and rephrased (page 9 and 10, first paragraph under statistical analysis).

9. Output of logistic regression is missing. What about model fitness, statistical significance of model, variation of dependent variables that can be explained from the independent variables etc.?

Authors’ response:

These details are provided in Table 4 (page 14) and Table 7 (page 17).

10. How independent variables were chosen in multivariable regression analysis?

Authors’ response:

Details about selecting independent variable in multivariate regression analysis is provided (page 10, first paragraph).

11. The term multiple in logistic regression is misnomer. It would have been multivariable logistic regression instead of multiple logistic regression.

Authors’ response:

Thanks for this note. This is corrected (page 10, first paragraph).

12. Table 1: Stress component: Age group & Job title – percentage is not coming to 100%. Please correct.

Authors’ response:

This occurred because of rounding. Now 2 decimals are provided for these (page 11, Table 1).

13. Table 3, 4, 6 & 7: Mention the chi-square statistic along with P value.

Authors’ response:

Chi square is mentioned now (pages 13-17, Tables 3, 4, 6, and 7).

14. Table 4, 6, 7: Mention years within parenthesis beside age group.

Authors’ response:

Years is mentioned now (pages 14-17, Tables 4, 6, and 7).

15. Table 4, 7: Footnote: Correct as ‘Crude odds ratio’ instead of ‘Crudes odds ratio’.

Authors’ response:

It is corrected (pages 14 and 17, Tables 4 and 7). 

16. Table 5: Item 1 – percentage is not coming to 100%. Also check the other percentages for any corrections.

Authors’ response:

This is because of rounding. Two decimals are now provided to correct this (page 15, Table 5).

17. Table 2 & 5: Better to mention the number and percentages as No. (%) format to have smooth look instead of keeping in separate columns.

Authors’ response:

The format of No. (%) is changes as suggested (pages 12 and 15, Tables 2 and 5).

Reviewer #2: 

1. Applying PSS and GAD in separate google forms and analyzing them together is problematic and lacks validity.

Authors’ response:

Thanks for this important note. This limitation is now highlighted and discussed (page 21, second paragraph).

2. While using a scale, referencing a disorder as yes or no according to Normal distribution is not acceptable and flawed. Rather it should be validated first and then to be diagnosed as yes or no according the to the cutoff marks.

Authors’ response:

Thank so much for this important comment. The part about normal distribution classification is removed and the distribution is now based on the tool guidelines and previous studies. References for such distribution are provided (Page 8, first and second paragraphs). 

3. It was written as 'The level of generalized anxiety disorder was measured based on the 7-item Generalized Anxiety Disorder scale.' GAD7 is a screening tool for GAD and can also be used for measuring Anxiety symptoms. So reframing this sentence(similar sentences are repeated many times in the manuscript) should be done.

Authors’ response:

This sentence and other similar sentences throughout the manuscript were reframed.

---

## [Decision Letter · Decision Letter 2]

25 May 2021

PONE-D-20-30168R2

Stress and anxiety among physicians during the COVID-19 outbreak in the Iraqi Kurdistan Region: An online survey

PLOS ONE

Dear Dr. Shabila,

Thank you for submitting your manuscript to PLOS ONE. After careful consideration, we feel that it has merit but does not fully meet PLOS ONE’s publication criteria as it currently stands. Therefore, we invite you to submit a revised version of the manuscript that addresses the points raised during the review process.

Authors must address all comments by reviewer 1. They must also provide clarification to comments 1,2,3,6, and 7 provided by reviewer 2. 

We look forward to receiving your revised manuscript.

Kind regards,

Arista Lahiri, MBBS, MD

Academic Editor

PLOS ONE

Journal Requirements:

Additional Editor Comments (if provided):

Authors must address all comments by reviewer 1. They must also provide clarification to comments 1,2,3,6, and 7 by reviewer 2. Authors should consider discussing about their study's overall response rate in their discussion section.

Reviewers' comments:

Reviewer's Responses to Questions

**Comments to the Author**

1. If the authors have adequately addressed your comments raised in a previous round of review and you feel that this manuscript is now acceptable for publication, you may indicate that here to bypass the “Comments to the Author” section, enter your conflict of interest statement in the “Confidential to Editor” section, and submit your "Accept" recommendation.

Reviewer #1: All comments have been addressed

Reviewer #2: (No Response)

2. Is the manuscript technically sound, and do the data support the conclusions?

Reviewer #1: Yes

Reviewer #2: No

3. Has the statistical analysis been performed appropriately and rigorously? 

Reviewer #1: Yes

Reviewer #2: No

4. Have the authors made all data underlying the findings in their manuscript fully available?

Reviewer #1: Yes

Reviewer #2: Yes

5. Is the manuscript presented in an intelligible fashion and written in standard English?

Reviewer #1: Yes

Reviewer #2: Yes

6. Review Comments to the Author

Reviewer #1: The article needs following clarifications:

The term ‘Multivariate’ should better be replaced by ‘Multivariable’.

Table 1: Two digits after decimals are not desirable while showing distribution of the study subjects. Again the uniformity will not be maintained. Thus please keep single digit after decimal and modify accordingly.

Reviewer #2: 1. Clarify the sample size calculation and precision level. Why chosen 3.5% as precision, though conventionally we take 5% as precsion. Non response rate how much to be taken here?

2. Translation of the questionnaire is important. Back translation and validation is required even for English speaking persons. Eventhough questions are having medical terminologies that just not justify the reasons of non medical persons not responding this question. Moreover PSS scale and GAD scale is used for Common people also. Restriction of response strictly for medical persons must be maintained

3. Exclusion criteria of study participants must be mentioned. Otherwise reasons of covid related stress and anxiety can never be established

4. Better to use the term univariate analysis because logistic regression is strictly for dichotomous outcome controlling confounding factors

5. Response rate 44% is unacceptable in generalised Anxiety disorder

6. In discussion part, writing justification of intervention should be modified. Counselling can be given..and covid protocol guidelines to be followed.

7. There are many lines in the discussion which are not related with the topic of interest, hence commenting these lines is absolute unnecessary and not acceptable. For example:

a. If mental health services are warranted, it

384 would be for the physicians showing a reverse correlation between stress and anxiety [37]. For

385 example, physicians with alexithymia in need of treatment might have high stress with low

386 anxiety. Those whose anxiety disorder has been tripped off by minimal stress may be

387 characterized by low stress and high anxiety.

b. The findings of the current study do not justify mental health interventions such as

352 counseling and the use of anxiolytic treatment or pharmacotherapy at this stage. - How authors came to this conclusion, please explain.

7. PLOS authors have the option to publish the peer review history of their article (what does this mean?). If published, this will include your full peer review and any attached files.

Reviewer #1: **Yes: **Indranil Saha

Reviewer #2: No

---

## [Author Response · Author response to Decision Letter 2]

30 May 2021

Thank you very much for sharing the academic editor's and reviewers' comments and suggestions on the manuscript entitled "Stress and anxiety among physicians during the COVID-19 outbreak in the Iraqi Kurdistan Region: An online survey." We would like to thank them for these valuable and helpful comments and suggestions, which helped in improving the manuscript's quality and clarity.

We have made the necessary revision by responding to the suggested comments. Please find below explanations to the revision made through a point-to-point response to the comments. The changes related to the Reviewer 1 comments are in red color font, while the changes related to Reviewer 2 comments are in the blue color font.

Journal Requirements:

Authors’ response:

All the references were checked again. We could not detect any retracted paper cited in our manuscript. We only found that reference number 38 in the previous version of this manuscript (Goodwin 2015) had a problem as the journal (Dialogues Clin Neurosci) is no longer being published, and the website has now been closed permanently. However, and following Reviewer 2 comment no. 7, this reference was removed as part of removing unnecessary and not acceptable sentences.

As part of revising the manuscript based on the academic editor and reviewers’ comments, two references were removed by removing the unnecessary sentences (references 37 and 38 from the previous version of the manuscript). Two new references were added when discussing the low response rate (references 37 and 38 in the current version of the manuscript).

Academic Editor Comments

Authors must address all comments by reviewer 1. They must also provide clarification to comments 1,2,3,6, and 7 by reviewer 2. Authors should consider discussing about their study's overall response rate in their discussion section.

Authors’ response:

All the comments of Reviewer 1 were addressed as explained under each comment below. All the comments of Reviewer 2 were addressed, and necessary clarifications were provided under each comment below. The overall response rate is now thoroughly discussed in the discussion section (Last paragraph of page 21 and beginning of page 22).

Reviewer #1: 

The term ‘Multivariate’ should better be replaced by ‘Multivariable’.

Authors’ response:

The term ‘Multivariate’ was replaced by ‘Multivariable’, as suggested (Page 10 under Statistical analysis and pages 14 and 17).

Table 1: Two digits after decimals are not desirable while showing distribution of the study subjects. Again the uniformity will not be maintained. Thus please keep single digit after decimal and modify accordingly.

Authors’ response:

All the two digit decimals were changed to a single digit, and uniformity of all numbers is now maintained (Table 5 on page 15, Table on pages 11 and 12).

Reviewer #2:

1. Clarify the sample size calculation and precision level. Why chosen 3.5% as precision, though conventionally we take 5% as precsion. Non response rate how much to be taken here? 

Authors’ response:

We have used a more conservative and lower precision than the conventional 5% for the sample size calculation as the expected prevalence of stress was estimated to be within the upper 30% level (71.5%). It is generally recommended that when the rate or prevalence is close to the 25-30% of the exterminates at both sides (i.e., 25-30% and lower or 70-75% or higher), a lower precision or margin of error should be used for sample size calculation. If we have used a 5% precision, then the required sample size would have been 248 instead of 417, which is close to the sample responded to the anxiety component of our study. The reason for using a lower precision is now included in the manuscript (end of page 6 and beginning of page 7).

2. Translation of the questionnaire is important. Back translation and validation is required even for English speaking persons. Eventhough questions are having medical terminologies that just not justify the reasons of non medical persons not responding this question. Moreover PSS scale and GAD scale is used for Common people also. Restriction of response strictly for medical persons must be maintained

Authors’ response:

In the manuscript, we have now explained that the original questionnaire was in the English language, including the anxiety and stress measurement scales. As we also administered the questionnaire in the English language, there was no need to translate it or back translate it to the Kurdish language. We have also explained that all the obtained data sets were checked, and it is confirmed that all respondents were strictly physicians (Page 7, end of the first paragraph).

3. Exclusion criteria of study participants must be mentioned. Otherwise reasons of covid related stress and anxiety can never be established

Authors’ response:

We have now included the main exclusion criteria of the study (Page 7, end of the first paragraph).

4. Better to use the term univariate analysis because logistic regression is strictly for dichotomous outcome controlling confounding factors

Authors’ response:

The term ‘univariate analysis' is now used throughout the manuscript (page 10 under the Statistical analysis section). 

5. Response rate 44% is unacceptable in generalised Anxiety disorder

Authors’ response:

The low overall response rate and its impact are now thoroughly discussed in the Discussion section (Last paragraph of page 21 and beginning of page 22).

6. In discussion part, writing justification of intervention should be modified. Counselling can be given..and covid protocol guidelines to be followed.

Authors’ response:

This paragraph was modified, and the need for intervention through counseling is emphasized (last paragraph of page 20, the conclusion section in the Abstract, and the main conclusion section on page 23).

7. There are many lines in the discussion which are not related with the topic of interest, hence commenting these lines is absolute unnecessary and not acceptable. For example:

a. If mental health services are warranted, it would be for the physicians showing a reverse correlation between stress and anxiety [37]. For example, physicians with alexithymia in need of treatment might have high stress with low anxiety. Those whose anxiety disorder has been tripped off by minimal stress may be characterized by low stress and high anxiety.

Authors’ response:

The discussion section was revisited, and the unnecessary lines and sentences were removed, including the above line and some other sentences (pages 21 and 22).

b. The findings of the current study do not justify mental health interventions such as counseling and the use of anxiolytic treatment or pharmacotherapy at this stage. - How authors came to this conclusion, please explain.

Authors’ response:

The above part of the discussion was revisited. We agree with the reviewer's comment that intervention is required, and we have included that in our first version of the submitted manuscript. However, the above sentence and non justification for mental health intervention were recommended by the previous academic editor or reviewer who initially handled this manuscript in the very first round of peer review. We have revisited the findings and the discussion section again and agree that interventions through counseling are needed at this stage based on the COVID-19 guidelines protocol. Thus, we have modified the above sentence and included the need for intervention in the manuscript (last paragraph of page 20, the conclusion section in the Abstract, and the main conclusion section on page 23).

---

## [Decision Letter · Decision Letter 3]

16 Jun 2021

Stress and anxiety among physicians during the COVID-19 outbreak in the Iraqi Kurdistan Region: An online survey

PONE-D-20-30168R3

Dear Dr. Shabila,

We’re pleased to inform you that your manuscript has been judged scientifically suitable for publication and will be formally accepted for publication once it meets all outstanding technical requirements.

Kind regards,

Arista Lahiri

Academic Editor

PLOS ONE

Additional Editor Comments (optional):

The revised version of the manuscript is now scientifically acceptable.

Reviewers' comments:

Reviewer's Responses to Questions

**Comments to the Author**

1. If the authors have adequately addressed your comments raised in a previous round of review and you feel that this manuscript is now acceptable for publication, you may indicate that here to bypass the “Comments to the Author” section, enter your conflict of interest statement in the “Confidential to Editor” section, and submit your "Accept" recommendation.

Reviewer #1: All comments have been addressed

Reviewer #2: All comments have been addressed

2. Is the manuscript technically sound, and do the data support the conclusions?

Reviewer #1: Yes

Reviewer #2: Yes

3. Has the statistical analysis been performed appropriately and rigorously? 

Reviewer #1: Yes

Reviewer #2: Yes

4. Have the authors made all data underlying the findings in their manuscript fully available?

Reviewer #1: Yes

Reviewer #2: Yes

5. Is the manuscript presented in an intelligible fashion and written in standard English?

Reviewer #1: Yes

Reviewer #2: Yes

6. Review Comments to the Author

Reviewer #1: All my comments are adequately addressed. Accepted from my part.

All things are adequately met from the authors' side

Reviewer #2: Authors addressed all the queries asked by me. Redundant lines in discussion sectioned pruned, Now it can be accepted.

7. PLOS authors have the option to publish the peer review history of their article (what does this mean?). If published, this will include your full peer review and any attached files.

Reviewer #1: **Yes: **Indranil Saha

Reviewer #2: **Yes: **Seshadri Sekhar Chatterjee

---

## [Editor Report · Acceptance letter]

21 Jun 2021

PONE-D-20-30168R3 

Stress and anxiety among physicians during the COVID-19 outbreak in the Iraqi Kurdistan Region: An online survey 

Dear Dr. Shabila:

I'm pleased to inform you that your manuscript has been deemed suitable for publication in PLOS ONE. Congratulations! Your manuscript is now with our production department. 

Kind regards, 

on behalf of

Dr. Arista Lahiri 

Academic Editor

PLOS ONE